# scMoMtF: An interpretable multitask learning framework for single-cell multi-omics data analysis

**Wei Lan** [1]*, **Tongsheng Ling**[1], **Qingfeng Chen**[1], **Ruiqing Zheng**[2], **Min Li**[2], **Yi Pan**[3]

**1** Guangxi Key Laboratory of Multimedia Communications and Network Technology, School of computer, electronic and information, Guangxi university, Nanning, Guangxi, China, **2** School of computer and engineering, Central South University, Changsha, Hunan, China, **3** School of Computer Science and Control Engineering, Shenzhen University of Advanced Technology, Shenzhen, Guangdong, China

* lanwei@gxu.edu.cn

## Abstract

With the rapidly development of biotechnology, it is now possible to obtain single-cell multi-omics data in the same cell. However, how to integrate and analyze these single-cell multi-omics data remains a great challenge. Herein, we introduce an interpretable multi-task framework (scMoMtF) for comprehensively analyzing single-cell multi-omics data. The scMoMtF can simultaneously solve multiple key tasks of single-cell multi-omics data including dimension reduction, cell classification and data simulation. The experimental results shows that scMoMtF outperforms current state-of-the-art algorithms on these tasks. In addition, scMoMtF has interpretability which allowing researchers to gain a reliable understanding of potential biological features and mechanisms in single-cell multi-omics data.

**Data Availability Statement:** scMoMtF is implemented by Python and the source code can be freely obtained at https://github.com/lanbiolab/scMoMtF. The datasets in this paper are all publicly

## Author summary

The rapidly developing single-cell multi-omics technologies enable the measurement of various modalities from the same cell. Integrative analysis of multi-modal data can provide new biological insights into the cellular state from different perspectives. However, this also poses challenges for the development of computational methods and tools for integrative analysis. We have developed a model called scMoMtF, which is capable of addressing multiple key tasks of single-cell multi-omics data analysis within a unified framework, including dimension reduction, cell classification and data simulation. Furthermore, scMoMtF is interpretable and can reveal potential marker genes and capture the complex relationships between single-cell multi-omics data.

## Introduction

The rapid development of single-cell sequencing technology makes it easier to analyze cell identity and behavior [1–4]. For example, the single-cell RNA sequencing (scRNA-seq) is

available. SNARE-seq, SHARE-seq and CITE-seq datasets are available from the GEO repository under the following accession codes: GSE126074, GSE140203 and GSE164378. PBMC dataset is available from 10X website (https://support.10xgenomics.com/single-cell-multiome-atac-gex/datasets/1.0.0/pbmc_granulocyte_sorted_10k). The preprocessed datasets is available at https://doi.org/10.5281/zenodo.13855396. The experimental data is available at https://doi.org/10.5281/zenodo.13843614.

**Funding:** This work was partially supported by the National Natural Science Foundation of China (No. 62472108 to W.L.; No. U24A20256 to W.L.; No. 62072122 to W.L.), the Natural Science Foundation of Guangxi (No. 2023JJG170006 to W.L.), the Guangxi BaGui Top Youth Talent Program to W.L, the Project of Guangxi Key Laboratory of Eye Health (No. GXYJK-202407 to W.L.), the Project of Guangxi Health Commission eye and related diseases artificial intelligence screen technology key laboratory (No. GXYAI-202402 to W.L.). The funders had no role in study design, data collection and analysis, decision to publish, or preparation of the manuscript.

**Competing interests:** The authors have declared that no competing interests exist.

widely used to measure the gene expression [5, 6] and the single-cell Assay for Transposase Accessible Chromatin with high-throughput (scATAC-seq) can measure chromatin accessibility [7]. However, these sequencing techniques only focus on the special molecular characteristics of single modality [8]. Therefore, the analysis of single-omics single-cell data only obtains partial information about the heterogeneity among various cells and fails to reveal the differences between cells [2].

The single-cell multi-omics technology can deconstruct the heterogeneity of cells within complex biological systems. For example, single-nucleus chromatin accessibility and mRNA expression sequencing (SNARE-seq) [9] and simultaneous high-throughput ATAC and RNA expression with sequencing (SHARE-seq) [10] techniques can measure gene expression and chromatin accessibility simultaneously in the same cell. In addition, cellular indexing of transcriptomes and epitopes by sequencing (CITE-seq) can measure single-cell gene expression and use the counting of antibody-derived tags (ADT) to quantify surface protein [11]. These single-cell omics data provide useful biological information from different views. Therefore, it is important to integrate these single-cell multi-omics data for obtaining a deeper biological understanding of cell [12–14].

Increasing methods based on deep learning have been proposed for analyzing single-cell multi-omics data such as dimension reduction, cell classification and data simulation. Dimension reduction is an important step in clustering analysis which can explore biological information at the cell type or subtype level. MultiVI [15] and totalVI [16] can obtain single-cell multi-omics joint embeddings by dimension reduction and perform cluster analysis by using some simple clustering algorithms. However, these methods only focus on joint embedding which prevents them from obtaining dimension reduction data that more conducive to cluster analysis. To address this issue, Lin et al. [17] propose an end-to-end deep learning model to learn the potential features of embedding for clustering analysis. In addition, cell classification task is also a key task in single-cell multi-omics data analysis. Many methods have been proposed recently for transferring cell type labels across modalities [18]. For example, Lin et al. [19] propose a scalable transfer learning method to annotate scATAC-seq data by using a large amount of high-quality annotated scRNA-seq data and Cao et al. [20] designed a cell label transmission strategy for single-cell multi-omics data based on coupled-VAE and Minibatch-UOT methods. However, few methods are designed for cell classification by using all modal data of single-cell multi-omics data. Currently, many methods which focus on cell classification only use single-omics data (such as scRNA-seq data). For example, Alquicira-Hernandez et al. [21] propose a method to classify cells in scRNA-seq data by combining unbiased feature selection from a reduced-dimension space and machine-learning probability-based prediction method and Lin et al. [22] propose a multiscale classification framework based on ensemble learning to classify cells of scRNA-seq data. For data simulation task, the goal is to increase the number of cells in sparse cell clusters to improve the quality of multi-omics single-cell data. For example, Liu et al. [23] propose a multi-tasking method (Matilda) to simulate single-cell multi-omics data. These methods have achieved a great success in single-cell multi-omics data. However, most of methods only focus on solving a single problem and require a lot of training time which makes it difficult to adapt the gradually growing needs of single-cell multi-omics data analysis. In addition, most methods increase the depth of the model in order to obtain stronger learning ability which makes it difficult to track the contribution of model inputs and loss the interpretability of model [24].

In this paper, we propose an interpretable multitask learning framework (scMoMtF) for single-cell multi-omics data analysis. scMoMtF can simultaneously solve multiple key tasks of single-cell multi-omics data analysis including dimension reduction, cell classification and data simulation. The shared information between tasks can be utilized to complement each

other to improve the learning ability of scMoMtF and the depth of scMoMtF can be reduced to ensure interpretability by using multitask learning. We evaluate the dimension reduction performance of scMoMtF on four different datasets from SNARE-seq, peripheral blood mononuclear cell (PBMC) [25], SHARE-seq and CITE-seq [26]. The experimental results indicate that dimension reduction data of scMoMtF can better distinguish cell subtypes and have higher clustering consistency than the current state-of-the-art methods. For cell classification task, we compare scMoMtF with four cutting-edge single-omics classification methods. The results of five-fold cross-validation show that scMoMtF has significantly higher accuracy in cell classification than other methods. In addition, scMoMtF can accurately simulate cells in different modalities. We also analyze the interpretability of scMoMtF and the results indicate that scMoMtF has the ability to reveal potential marker genes and capture complex relationships between single-cell multi-omics data [27]. Finally, we demonstrate that scMoMtF can correct batch effect and requires shorter training time than other methods.

## Results

### The scMoMtF model

scMoMtF is composed of encoder module, decoder module, discriminator module and classification module (Fig 1A). Two independent modal encoders are designed to obtain embedding which contain key biological information from two modalities of single-cell multi-omics data (RNA, ATAC/ADT), respectively. Then, the two embeddings are concatenated and the concatenated embedding are input to cell encoder to obtain the final cell embedding. Further, the reconstructed data is obtained from cell embedding by using two independent modal decoders. In addition, the classification module is utilized to classify cell types based on final cell embedding. Finally, the discriminator module is designed to against the generator module which consist of the encoder and decoder module [28]. scMoMtF can complete three important tasks simultaneously. For example, the encoder module of scMoMtF can achieve dimension reduction for single-cell multi-omics data, the classification module of scMoMtF allows for accurate cell classification by using the encoded cell embedding and the generator module can simulate the data which input into the model (Fig 1B). In addition, the interpretability module is used to provide additional insights on the importance of genes in dimension reduction task and cell classification task. This helps to discover potential marker genes in the cell (Fig 1C).

### Performance on single-cell multi-omics data dimension reduction

To evaluate the performance of dimension reduction task on single-cell multi-omics data, we compare scMoMtF with current popular methods including MultiVI [15], totalVI [16], scMDC [17] and Matilda [23]. In these method, MultiVI [15] are designed for RNA modality and ATAC modality. totalVI [16] are designed for RNA modality and ADT modality. scMDC [17] and Matilda [23] are designed for RNA and ATAC/ADT modalities. We set the dimension of the biological information vector of each modality as 150, which is obtained by modality encoder. In addition, the dimension of cell embedding is set to 64 by using cell encoder. For all comparison methods, we use their default dimension for experiments. It should be noted that the donor 2 of CITE dataset and all data of other three datasets are selected as experimental data. We visualize the cell embedding of each model by using uniform manifold approximation and projection (UMAP) (Fig 2A–2D). It can be found the cell embedding of scMoMtF can provide clearer division between different cell clusters, especially for small numbers of cell subtypes. For example, scMoMtF can clearly separate the three cell subtypes (B intermediate, B memory and B naive), while the other methods can not exhibit clear cell

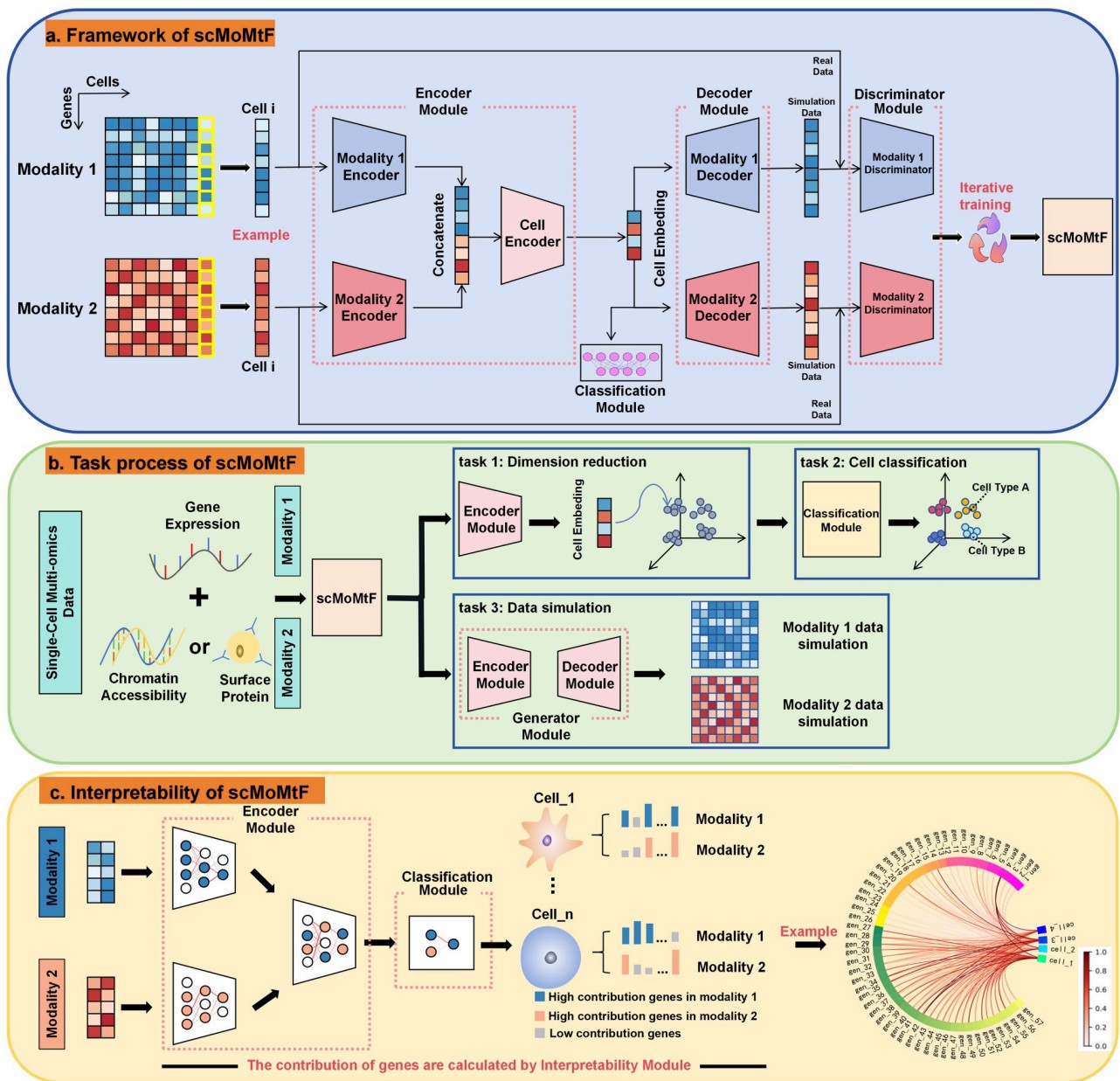

**Fig 1. scMoMtF overall structure and task module diagram. A** scMoMtF uses the matched single-cell multi-omics data as the input to the model and the overall model framework is encoder-decoder-discriminator-classifier. **B** The tasks process of scMoMtF. **C** The research process for the interpretability of scMoMtF.

cluster boundaries in CITE-seq dataset. In order to intuitively show the dimension reduction performance of each method, we use k-means clustering algorithm to cluster the cell embedding with same parameters (n_clusters is the number of cell types for the corresponding dataset and n_init is set to 30). We use three quantitative metrics including adjusted mutual information (AMI), normalized mutual information (NMI) and adjusted rand index (ARI) by five-fold cross-validation to measure the cluster performance [29–31]. It can be found that scMoMtF achieves higher AMI, NMI and ARI scores in different datasets (Fig 2E–2H). For example, in PBMC dataset, the AMI, NMI and ARI scores of scMoMtF are 0.847, 0.852, 0.740,

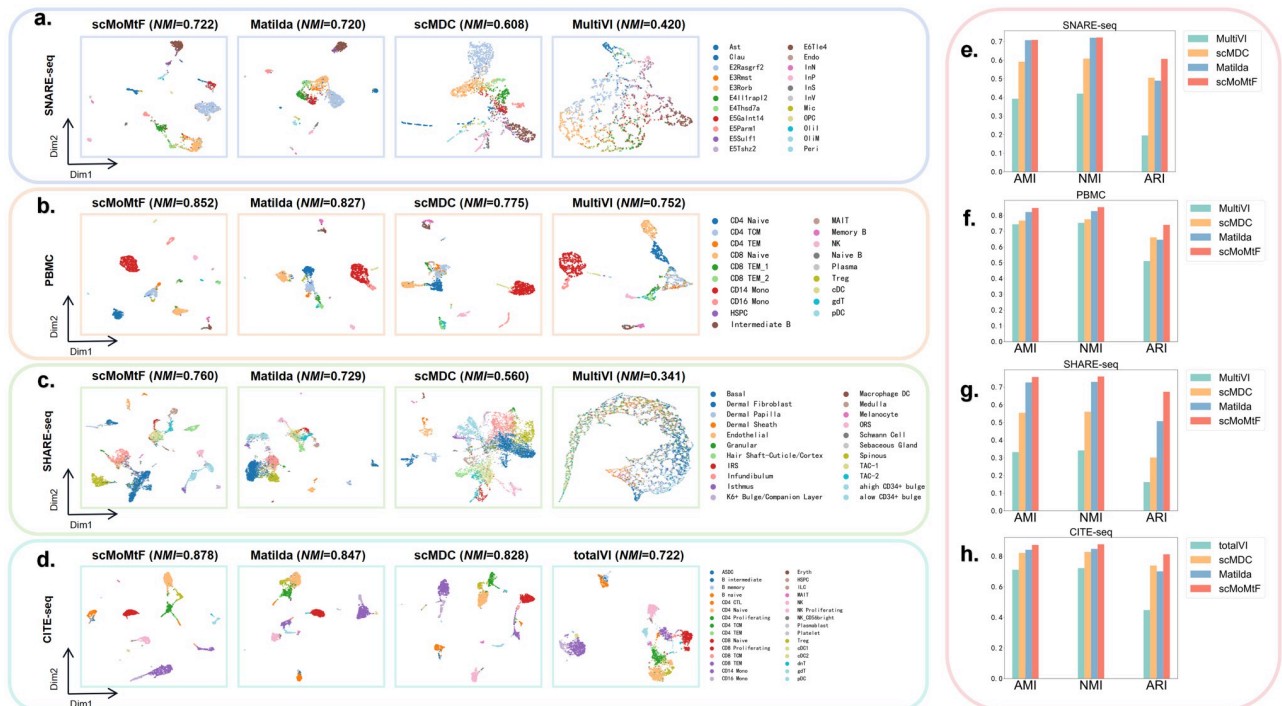

**Fig 2. Visualization and performance evaluation of dimension reduction task of scMoMtF compared with other comparison algorithms. A-C** Visualization of dimension reduction data generated by scMoMtF, Matilda, scMDC, and MultiVI on SNARE-seq, PBMC, and SHARE-seq datasets. **D** Visualization of dimension reduction data generated by scMoMtF, Matilda, scMDC and totalVI on the CITE-seq dataset. **E-G** Evaluate the clustering performance of dimension reduction data generated by scMoMtF, Matilda, scMDC, and MultiVI on SNARE-seq, PBMC, and SHARE-seq datasets using AMI, NMI, and ARI. **H** The clustering performance of dimension reduction data generated by scMoMtF, Matilda, scMDC and totalVI on CITE-seq dataset.

which outperforms other methods (MultiVI: 0.743, 0.752, 0.510; scMDC: 0.767, 0.775, 0.660; Matilda: 0.821, 0.827, 0.645). In addition, although the performance of Matilda is close to scMoMtF in the SNARE-seq dataset, scMoMtF performs more stable in the other datasets. These experimental results demonstrate the superior performance of scMoMtF in dimension reduction task for single-cell multi-omics data.

## Performance on single-cell multi-omics data cell classification

Previous methods focus on cell label transmission between different data modalities. There are few methods for cell type classification task by using all single-cell multi-omics data together. In order to prove that scMoMtF has better performance and robustness in classifying cell types by using single-cell multi-omics data, we compare scMoMtF with the state-of-the-art methods for cell type classification based on RNA modality including scPred [21], scClassify [22], scmap [32] and CHETAH [33]. We also use five-fold cross-validation to evaluate the classification accuracy. It can be observed that scMoMtF has a higher classification accuracy on these datasets than other methods which only with RNA modality (Fig 3A). It should be noted that the classification accuracy of scMoMtF are all over 84% on these datasets and this reflects the robustness of scMoMtF to different single-cell data. In addition, it also can be found that scMoMtF is able to correctly classify rare cells in these datasets. For example, comparing with scPred [21] which is the second best model in performance. scMoMtF achieves better classification performance on rare cells (the cell types that have small proportion in the dataset) such as Plasma (0.1% in the dataset), Treg (1.6% in the dataset) and gdT (1.4% in the dataset) (Fig

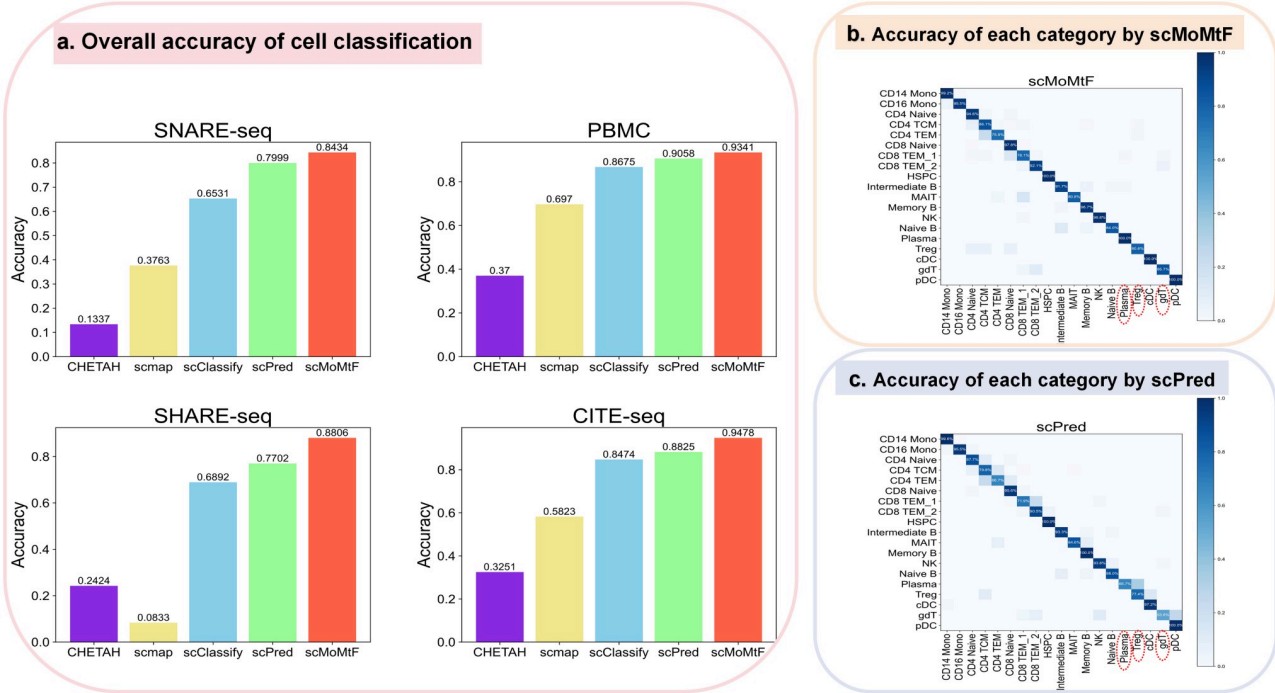

**Fig 3. Cell classification performance of scMoMtF. A** Comparison of classification accuracy between scMoMtF and other comparison algorithms under five-fold cross-validation. **B** The classification results of scMoMtF for each cell type in the PBMC dataset. **C** The classification results of scPred for each cell type in the PBMC dataset.

3B and 3C). The classification accuracy of scMoMtF for Plasma, Treg and gdT is 100%, 80.6% and 85.7%, respectively. The classification accuracy of scPred for Plasma, Treg, and gdT is 66.7%, 77.4% and 53.6%. These results demonstrates that scMoMtF improves the classification accuracy of rare cells which contributes to whole performances improvement of cell classification on different datasets.

## Performance on single-cell multi-omics data simulation

There are two tasks on the single-cell multi-omics data simulation: specific cell type data simulation and multiple cell types data simulation. For specific cell type data simulation task, we apply scMoMtF to the PBMC and CITE-seq datasets. In the PBMC dataset, we use the generator of the trained model to simulate 200 CD14 Mono cells. Then, we use UMAP to visualize CD14 Mono cells of real data and simulated data in the RNA modality and ATAC modality, respectively. It can be seen that there is almost no difference between the real data represented by the red dots and the simulated data represented by the blue dots (Fig 4A and 4B). In addition, it can be found that the simulated data generated by scMoMtF can eliminate outliers in the real data (Fig 4A). This result shows that scMoMtF is able to accurately simulate the CD14 Mono cells of real data in both RNA modality and ATAC modality. In the CITE-seq dataset, we simulate NK cells and visualize NK cells of real data and simulated data in ADT modality (Fig 4C). It can be observed that NK cells of simulated data and real data are highly similar. The experimental results show that scMoMtF can simulate single-cell multi-omics data well with specific cell types. For the multiple cell types data simulation task, we select top-100 highly variable genes (HVGs) in both the real data and the simulated data and calculate the pearson correlation of HVGs between the real data and simulated data on the SNARE-seq, PBMC,

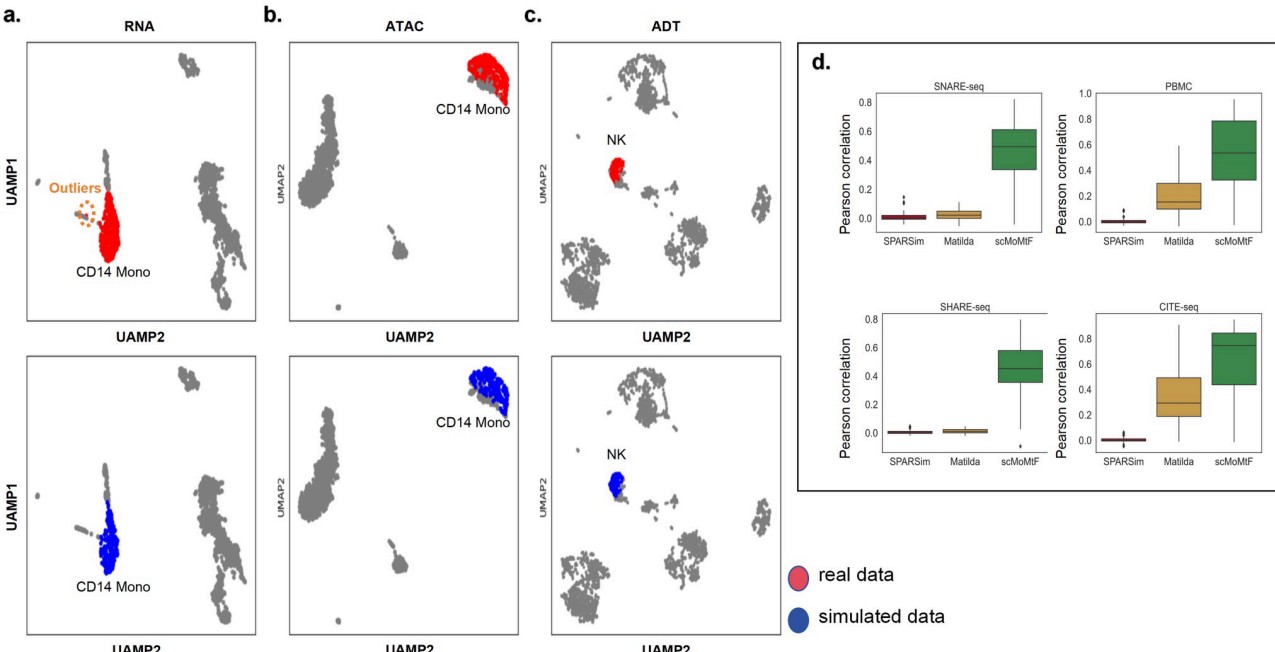

**Fig 4. scMoMtF single-cell multi-omics data simulation performance. A-B** scMoMtF visualizes the simulation effects of specified cell types on PBMC datasets. **C** scMoMtF visualizes the simulation effects of specified cell types on the CITE-seq dataset. **D** Pearson's correlation between scMoMtF and other single-cell data simulation methods for highly variable genes in real and simulated data. Lower and upper hinges, first and third quartiles(Q1,Q3); whiskers, range of 1.5-times the interquartile; Centre line, median; Dot, outliers.

SHARE-seq and CITE-seq datasets. We compare scMoMtF with Matilda [23] and SPARsim [34] on RNA modality. It can be seen that scMoMtF achieves higher pearson correlation between real data and simulated data on four datasets (Fig 4D). This indicates that scMoMtF makes the correlation structures between real data and simulated data more similar than other methods in multiple cell types data simulation task. In summary, scMoMtF has a good effect on single-cell multi-omics data simulation tasks.

## scMoMtF corrects batch effects

For single-cell multi-omics data, the batch effect mask true biological variation which may obtian the unreliable analysis result [35]. Therefore, it needs to correct batch effect in single-cell multi-omics data analysis [36]. In order to demonstrate the performance of scMoMtF in correcting batch effects, we selecte the first five donors out of the eight donors as five batch data (P1,2,3,4,5) in the CITE-seq dataset and use UMAP to visualize the raw data. It can be found that there is a serious batch effect in the raw data and cells tended to cluster by donor rather than by cell type (Fig 5A). We train scMoMtF on individual batch as reference data to correct the remaining batches. It can be observed that scMoMtF can effectively correct batch effects to make cells of the same type gather well together (Fig 5B). In addition, for evaluating the performance of cell classification across batch, we use each batch as training data and other remaining batches as test data. It should be noted that the average classification accuracy of other batches is used as final classification result for each batch. The results show that scMoMtF can obtain more than 90% classification accuracy across batches with almost no fluctuation (Fig 5C). In summary, scMoMtF can be used to solve the batch effect problem to obtain more reliable results in single-cell multi-omics data.

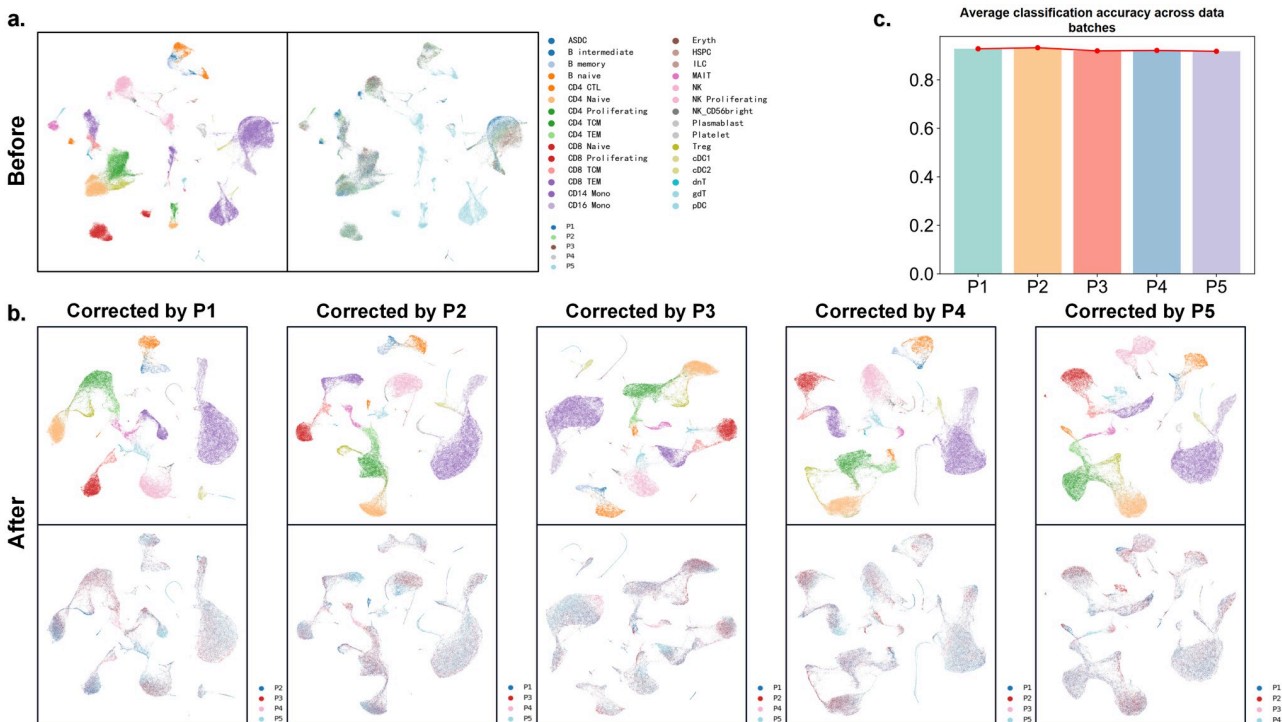

**Fig 5. scMoMtF corrects batch effect in the CITE-seq dataset. A** Visualization of selected data before removing batch effect. **B** Visualization of the results of batch correction by scMoMt using different batches. **C** The average classification accuracy across data batches of different batches.

## The interpretability of scMoMtF

In order to show the interpretability of scMoMtF, we use SHapley Additive exPlanation (SHAP) [37] to analyze the model. The core idea of SHAP is to calculate the marginal contribution of features of the model. We embed SHAP into our interpretability module. Herein, we analyze scMoMtF on the PBMC dataset by interpretability module. We visualize the data in RNA modality and ATAC modality, respectively (Fig 6A). Among all cell types, we focus on CD8+ T cells which contians three cell subtypes including CD8 Naive, CD8 TEM_1 and CD8 TEM_2. In addition, the peaks of ATAC data in the PBMC dataset are mapped to corresponding genes. Then, the interpretability module is used to calculate the important scores of genes of RNA modality and ATAC modality for dimension reduction and cell classification tasks. And top rank important genes are selected based on important scores.

In the RNA modality, it can be found that genes CD8B, CCL5 and GZMK with significant different contribution to the three cell subtypes (Fig 6B). The gene CD8B provides more contribution to subtype CD8 Naive than the other two subtypes. The gene CCL5 provides more contribution to subtypes CD8 TEM_1 and CD8 TEM_2. The gene GZMK provides more contribution to subtype CD8 TEM_1. In addition, we visualize these genes in their corresponding modalities. It can be seen that these genes with higher expression to their corresponding high contributing cell subtypes (Fig 6C). It indicates that these genes are cell-specific genes of CD8+ T cells and play an critical role in CD8+ T cells functions. The results can be proved by the previous study that the marker genes of CD8+ T cells contain CD8B and GZMK [38]. And it has been demonstrated that the low expression of gene CCL5 decreases the number of CD8+ T cells in cancer cells [39].

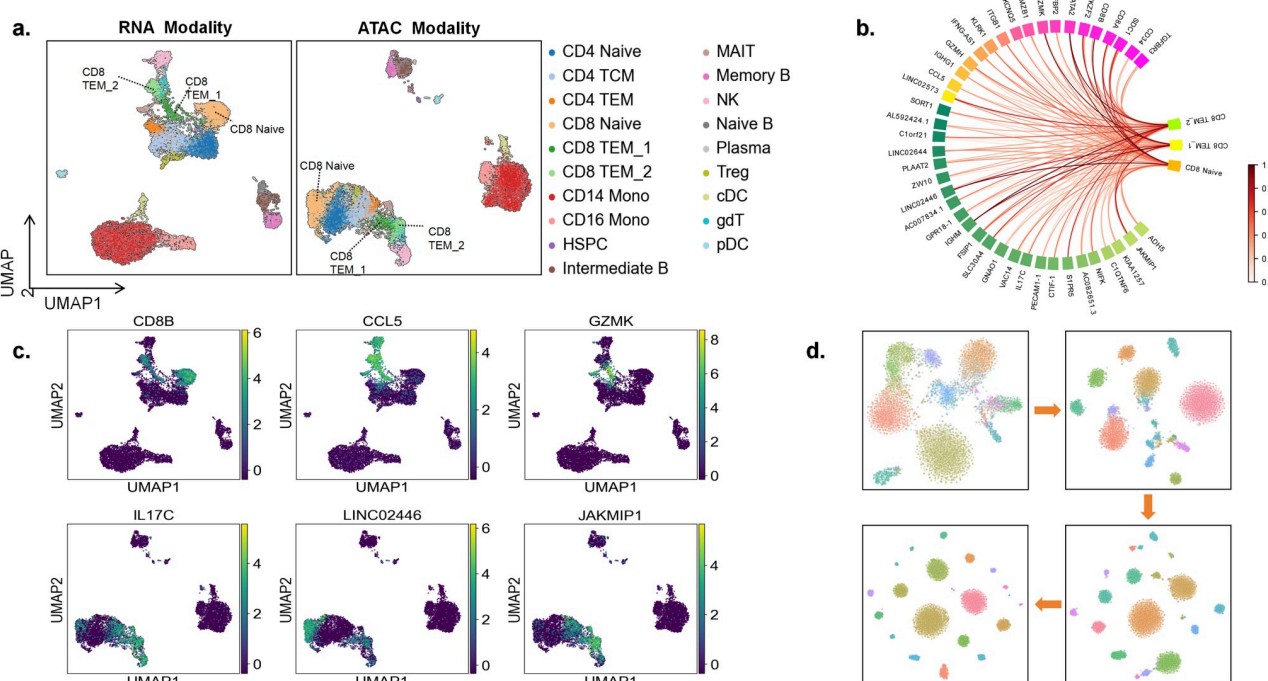

**Fig 6. Interpretability analysis diagram of scMoMtF. A** Visualize the location of the cell subtype of CD8+ T cells in both the RNA and ATAC modalities. **B** The characteristics with high contribution in CD8+ T cells are normalized to a value between 0 and 1. **C** The expression degree of CD8B, CCL5 and GZMK in RNA modality and the expression degree of IL17C, LINC02446 and JAKMIP1 in ATAC modality. **D** Visualize the cell embedding of scMoMtF at different training periods using t-SNE.

In the ATAC modality, genes IL17C, LINC02446 and JAKMIP1 also with great different contribution (Fig 6B). Similarly, we visualize the expression degree of these genes. The results (Fig 6C) show that these genes are also important genes of CD8+ T cells. It is proved by the latest research that the LINC02446 enhances IL7R abundance which leads to increase the proportion of Treg cells to promote melanoma metastasis and Treg cells are driven by CD8+ T cells which indicates that the increase of Treg cells will lead to simultaneously increase CD8+ T cells [40]. In addition, it has been domanstrated the JAKMIP1 may regulate CD8+ T cell infiltration by leukocyte migration, DCs, and T-cell recruitment [41].

Moreover, we use the t-SNE to visualize the cell embeddings of single-cell multi-omics data at different stages of training. It can be observed that cells gradually gather together and cell clustering becomes more pronounced with the training progresses (Fig 6D). The above experiments show that scMoMtF has reliable interpretability to help us reveal potential marker genes and can effectively capture complex relationships to obtain better cell embedding during training in single-cell multi-omics data.

## Training efficiency of scMoMtF

In the field of single-cell multi-omics data analysis, the performance and training efficiency of deep learning models are important criteria for evaluating their superiority. We record the runtime of all models in the experiment and the results are shown in Table 1. It can be found that scMoMtF has a significantly shorter training time by compared to other models (including both multi-task and single-task models). Although the training time of scmap [32] is shorter than scMoMtF, the accuracy of scmap in the cell classification task is much lower than

**Table 1. Task training time (in seconds) of each method on different datasets.**

| Task | Method | SNARE-seq (9190 Cells) | PBMC (9631 Cells) | SHARE-seq (17115 Cells) | CITE-seq (32231 Cells) |
|---|---|---|---|---|---|
| Dimension Reduction | MultiVI | 613 | 782 | 2269 | - |
| | totalVI | - | - | - | 2039 |
| | scMDC | 313 | 324 | 417 | 1033 |
| Cell Classification | scPred | 382 | 305 | 1260 | 4106 |
| | scClassify | 29 | 37 | 157 | 106 |
| | scmap | 3 | 5 | 15 | 9 |
| | CHETAH | 39 | 22 | 81 | 84 |
| Data Simulation | SPARSim | 152 | 180 | 279 | 587 |
| Multiple Tasks | Matilda | 40 | 42 | 67 | 143 |
| | scMoMtF | **14** | **14** | **28** | **46** |

**Note**: Among all the models scMoMtF and Matilda are multi-task models and the rest are single-task models. - : indicates that the model cannot be applied to the dataset.

scMoMtF. Therefore, scMoMtF not only demonstrates superior performance in multitasking capabilities but also exhibits exceptional competitiveness in training efficiency. And scMoMtF is a powerful tool for efficiently handling single-cell multi-omics data.

## Discussion

The current single-cell sequencing technology can simultaneously measure multiple molecular information (RNA, chromatin accessibility and proteins) of the same cell. It demand to combine different tasks to fully understand these single-cell multi-omics data. However, many current methods for analyzing single-cell multi-omics data are designed to perform a single task and rely on specific datasets which make it fail to fully utilize the potential of single-cell multi-omics data. For example, scMDC performs well on PBMC and CITE-seq datasets but performs poorly on other datasets in dimension reduction task. And the accuracy of scmap is significantly lower on the SHARE-seq dataset in cell classification task. In addition, many methods lack corresponding interpretability which is difficult to provide biologically reliable insights. To address this issue, we propose an interpretable multitask framework (scMoMtF) for comprehensive analyzing single-cell multi-omics data. We evaluate the performance of scMoMtF in data dimension reduction, cell classification and data simulation tasks. The experimental results indicate that scMoMtF can obtain better performance on all tasks and correct the batch effect of single-cell multi-omics data. In addition, scMoMtF can reveal potential marker genes to provide reliable biological insights. Furthermore, scMoMtF can be a convenient analysis tool without too much parameters adjustment and training time.

In future work, we also plan to explore potential improvements to the method, such as enhancing its computational efficiency to handle larger datasets more effectively and expanding its applicability to a broader range of single-cell multi-omics datasets. Moreover, we will investigate potential applications of scMoMtF in related areas, such as integrating spatial transcriptomics data or applying the framework to other types of multi-modal data.

## Materials and methods

### Overview of scMoMtF

The scMoMtF is a neural network model that can perform multiple single-cell multi-omics tasks. The scMoMtF consists of an encoder module, a decoder module, a discriminator

module and a classification module. We suppose that $X^{(m)} \in R^{n \times v^{(m)}}$ ($m = 1, \ldots, M$) represents single-cell data from modality $M$, where $n$ represents the number of cells and $v^{(m)}$ represents the number of features in $X^{(m)}$. In addition, $M$ is equal to 2 in this paper.

## The encoder module of scMoMtF

In the encoder module, we design two independent modality encoders $E^{(1)}_{Modality}$ and $E^{(2)}_{Modality}$ for different modalities, where $E^{(1)}_{Modality}$ encodes the data from modality 1 and $E^{(2)}_{Modality}$ encodes the data from modality 2. The each modal data in cell $i$ ($i = 1, \ldots, n$) is mapped to specific modal embedding $h^{(m)}_i$ for important multi-omics information extraction:

$$h^{(1)}_i = E^{(1)}_{Modality}(x^{(1)}_i) \tag{1}$$

$$h^{(2)}_i = E^{(2)}_{Modality}(x^{(2)}_i) \tag{2}$$

where $x^{(1)}_i$ is a row of $X^{(1)}$ denotes the data of cell $i$ from modality 1 and $x^{(2)}_i$ is a row of $X^{(2)}$ denotes the data of cell $i$ from modality 2. Next, $h^{(1)}_i$ and $h^{(2)}_i$ are concatenated to input into the cell encoder $E_{Cell}$ to obtain the final cell embedding $z_i$ of cell $i$:

$$z_i = E_{Cell}(concatenate(h^{(1)}_i, h^{(2)}_i)) \tag{3}$$

where the length of $h^{(1)}_i$ and $h^{(2)}_i$ are $l^{(1)}_i$ and $l^{(2)}_i$, respectively. And the length of concatenated embedding is $l^{(1)}_i + l^{(2)}_i$.

## The decoder module of scMoMtF

In the decoder module, we use two decoders $D^{(1)}_{Modality}$ and $D^{(2)}_{Modality}$ to reconstruct $z_i$ to the original feature dimensions of each modal data:

$$\hat{x}^{(1)}_i = D^{(1)}_{Modality}(z_i) \tag{4}$$

$$\hat{x}^{(2)}_i = D^{(2)}_{Modality}(z_i) \tag{5}$$

where $\hat{x}^{(1)}_i$ is reconstructed data of $x^{(1)}_i$ and $\hat{x}^{(2)}_i$ is reconstructed data of $x^{(2)}_i$.

## The discriminator module of scMoMtF

In scMoMtF, we treat the encoder module and decoder module as a single-cell multi-omics data generator. The discriminator module assists the generator generate data that is more similar to the original data. We design $Dis^{(m)}$ as a discriminator of modality $M$, and the input of the discriminator $Dis^{(m)}$ is $\hat{x}^{(m)}_i$ which is generated by using generator and raw data $x^{(m)}_i$. The purpose of $Dis^{(m)}$ is to achieve binary classification, and the result is the probability that the input data comes from a real data (as opposed to fake data).

## The classification module of scMoMtF

We input $z_i$ into a fully connected network to obtain a cell label vector $y_i$ with a length of $C$ ($C$ is the number of cell types in the input data) for cell $i$. The $y^{(c)}_i$ ($c = 1, 2, \ldots, C$) represents the

probability of cell $i$ is predicted as the $c$ class:

$$y_i = layer(z_i) \tag{6}$$

where *layer* is fully connected network.

## Reconstruction loss

The original data is mapped to the low dimensional common embedding space based on encoder module, and reconstructed to the original dimension based on the decoder module. The reconstruction loss is defined as:

$$L_{res} = \frac{1}{nM} \sum_{i=1}^{n} \sum_{m=1}^{M} \left\| \hat{x}_i^{(m)} - x_i^{(m)} \right\|_{2'} \tag{7}$$

$L_{res}$ is used to measure the distance between the original data and the reconstructed data.

## Classification loss

We use LSR (Label Smoothing Regularization) [42] to improve cross entropy loss function. We replace the real label vector $y_{real}$ with the updated label vector $y_{ls}$ based on label smoothing method:

$$y_{ls} = (1 - \alpha) \times y_{real} + \alpha/C \tag{8}$$

where $\alpha$ is a hyperparameter. Therefore, the cross entropy loss can be rewritten as follow:

$$L_{cls} = -\sum_{c=1}^{C} y_{ls}^{(c)} \log y_i^{(c)} \tag{9}$$

## Generator loss

We use the least square loss [43] as the loss function to train the generator. The generator loss is defined as follow:

$$L_{gen} = \frac{1}{nM} \sum_{i=1}^{n} \sum_{m=1}^{M} \| Dis^{(m)}(\hat{x}_i^{(m)}) - 1 \|_{2'}^{2} \tag{10}$$

$L_{gen}$ is to make the simulated data generated by the generator similar to the original data to the discriminator.

## Discriminator loss

We also use the least square loss as the loss function for the discriminator. The discriminator loss is defined as follow:

$$L_{dis} = \frac{1}{nM} \sum_{i=1}^{n} \sum_{m=1}^{M} \left\| Dis^{(m)}(\hat{x}_i^{(m)}) \right\|_{2}^{2} + \frac{1}{nM} \sum_{i=1}^{n} \sum_{m=1}^{M} \left\| Dis^{(m)}(x_i^{(m)}) - 1 \right\|_{2'}^{2} \tag{11}$$

$L_{dis}$ is to make the discriminator predict the simulated data as fake and the original data as true.

## scMoMtF training

For all datasets, we normalize the original count matrix by using scanpy to select the top 4000 highly variable genes for RNA modality, using episcanpy to select the top 4000 highly variable peaks for ATAC modality and preserving all features in ADT modality. Subsequently, the pre-processed data is input into the model for training, and the overall loss function during the training process is defined as follow:

$$L_{total} = L_{res} + \gamma \times L_{cls} + L_{gen} + L_{dis} \tag{12}$$

where $\gamma$ is a hyperparameter to control the influence of the classification module. We train scMoMtF on all experimental datasets and update each module to determine the optimal hyperparameter based on the loss function.

## Description of the dataset

The datasets used in the experiment are mainly matched datasets which contain matched RNA and ATAC/ADT data. There are four datasets used in the experiment:

**SNARE-seq dataset.** The original RNA and ATAC count matrices are measured from the mouse cerebral cortex by Chen et al. [9] and can be downloaded from the GEO website (accession code GSE126074). SNARE-seq contain matched RNA and ATAC data. We follow the processing steps of Lin et al. [19] for this dataset and obtain the pre-processed data. It consists of 9190 cells with 241757 features in ATAC and 28930 genes in RNA whit 22 cell types.

**PBMC dataset.** The 10x-Multiome-Pbmc10k dataset is downloaded from the 10 xgenomics [25] to obtain original gene expression and chromatin accessibility. We download this dataset from the preprocessed data provided by Cao et al. [44]. It consists of 9631 cells with 107194 features in ATAC and 29095 genes in RNA with 19 cell types.

**SHARE-seq dataset.** This dataset measures gene expression and chromatin accessibility in the same single-cell in mouse skin samples which is derived from Ma et al. [10]. The raw data is available to download from the GEO website (accession code GSE140203). The gene activity score matrix is obtained by Seurat [26], and cells with less than 1% gene expression are filtered out. It consists of 32231 cells with 340341 features in ATAC and 21478 genes in RNA with 22 cell types.

**CITE-seq dataset.** The raw data of this dataset is downloaded from the GEO website (accession code GSE164378) and provided by Hao et al. [26]. We download a preprocessing file of this dataset provided by Lakkis et al. [45] and remove cells labeled as Doublet from the cell type. This dataset consists of 161159 cells with 224 proteins in ATAC and 20729 genes in RNA from eight donors, which is treated as eight batches. And it has three cell type resolutions: L1 (8 types), L2 (30 types) and L3 (57 types). L1, L2 and L3 represent different levels of cell type resolution, L1 represents coarse-grained division of cell types, L2 and L3 represent higher-resolution subpopulation division. We only use L2 (30 types) in our experiment.

## Dimension reduction methods

**MultiVI (https://github.com/scverse/scvi-tools).** The input of MultiVI are matched raw count matrices of RNA and gene activity score matrices from ATAC. We use the default parameters in the experiment. Following the author's tutorial, we first connect the RNA and ATAC data and then train the model through the 'scvi.model.MULTIVI.setup_anndata', 'scvi.model.MULTIVI' and 'train' functions. The final embedding can be obtained by the 'get_latent_representation' function.

**totalVI (https://github.com/scverse/scvi-tools).**   The input of totalVI are matched raw count matrices of RNA and ADT. We use the default parameters in the experiment. Following the author's tutorial, we normalize the raw data through the 'normalize_total' and 'log1p' functions. And then we train the model through the 'scvi.model.TOTALVI.setup_anndata', 'scvi.model.TOTALVI' and 'train' functions. The final embedding can be obtained by the 'get_latent_representation' function.

**scMDC (https://github.com/xianglin226/scMDC).**   There are two types inputs of scMDC: matched raw count matrices of RNA and gene activity score matrices from ATAC; matched raw count matrices of RNA and ADT. We use the default parameters in the experiment. Following the author's tutorial, we normalize the raw data through the 'normalize' function. And then we train the model through the 'scMultiCluster' and 'pretrain_autoencoder' functions. The final embedding can be obtained by the 'encodeBatch' function.

**Matilda (https://github.com/PYangLab/Matilda).**   There are two types inputs of Matilda: matched raw count matrices of RNA and gene activity score matrices from ATAC; matched raw count matrices of RNA and ADT. We use the default parameters in the experiment. Following the author's tutorial, we normalize the raw data through the 'compute_log2' and 'compute_zscore' functions. Then we concatenate the data of the two modalities and train the model through the 'CiteAutoencoder_SHAREseq' (or 'CiteAutoencoder_CITEseq') and 'train_model' functions. The final embedding can be obtained by the 'get_encodings' function.

## Cell classification methods

**scPred (https://github.com/powellgenomicslab/scPred).**   The input of scPred is raw count matrices of RNA. We use the default parameters in the experiment. Following the author's tutorial, we preprocess the raw data through the 'NormalizeData', 'FindVariableFeatures', 'ScaleData', 'RunPCA' and 'RunUMAP' functions. And then we train the model through the 'getFeatureSpace' and 'trainModel' functions. The result of cell classification can be obtained by the 'scPredict' function.

**scClassify (https://github.com/SydneyBioX/scClassify).**   The input of scClassify is raw count matrices of RNA. We use the default parameters in the experiment. Following the author's tutorial, we normalize the raw data through the 'NormalizeData' function. And then we train the model and obtain the result of cell classification through the 'scClassify' function.

**scmap (https://github.com/hemberg-lab/scmap).**   The input of scmap is raw count matrices of RNA. We use the default parameters in the experiment. Following the author's tutorial, we train the model and obtain the result of cell classification through the 'selectFeatures', 'indexCluster' and 'scmapCluster' functions.

**CHETAH (https://github.com/jdekanter/CHETAH).**   The input of CHETAH is raw count matrices of RNA. We use the default parameters in the experiment. Following the author's tutorial, we train the model and obtain the result of cell classification through the 'CHETAHclassifier' function.

## Data simulation methods

**SPARsim (https://gitlab.com/sysbiobig/sparsim).**   The input of SPARsim is raw count matrices of RNA. Following the author's tutorial, we normalize the raw data through the 'scran_normalization' function. The parameters of SPARsim are estimated by 'SPARSim_estimate_parameter_from_data' function. And then we train the model and generate simulated data through the 'SPARSim_simulation' function.

**Matilda.**   The detailed information of Matilda can be seen in 'Dimension reduction methods' section. Matilda can generate simulated data of two modalities. After the Matilda is

trained, we use the function 'get_vae_simulated_data_from_sampling' to generate simulated data. And then we select the simulated data of RNA from the result.

### Evaluation metrics

**Adjusted Rand Index (ARI).** The ARI score measures measures the agreements between two sets $P$ (the clustering result of the predicted by model) and $T$ (the clustering result of real label). Assuming $N_1$ represent the number of pairs of objects that are assigned to the same cluster in both $P$ and $T$; $N_2$ represent the number of pairs of objects that are assigned to different clusters in both $P$ and $T$; $N_3$ represent the number of pairs of objects that are assigned to the same cluster in $P$ but to different clusters in $T$; $N_4$ represent the number of pairs of objects that are assigned to the same cluster in $T$ but to different clusters in $P$. the ARI is calculated using the following formula:

$$ARI = \frac{\binom{n}{2}(N_1 + N_2) - [(N_1 + N_3)(N_1 + N_4) + (N_4 + N_2)(N_3 + N_2)]}{\binom{n}{2} - [(N_1 + N_3)(N_1 + N_4) + (N_4 + N_2)(N_3 + N_2)]} \tag{13}$$

And the ARI is near one when the clustering result from the model aligns well with the observed cell type labels, while it is close to zero when the clustering resembles a random assignment.

**Normalized mutual information (NMI).** Similar to ARI score, let $P = \{P_1, P_2, \ldots, P_{np}\}$ and $T = \{T_1, T_2, \ldots, T_{nt}\}$ be the predicted and real labels on a dataset with $n$ cells. NMI is defined as follows:

$$NMI = \frac{I(P, T)}{\max\{H(P), H(T)\}} \tag{14}$$

$$I(P, T) = \sum_{i=1}^{np}\sum_{j=1}^{nt}|P_i \bigcap T_j|\log\frac{n|P_i \cap T_j|}{|P_i| \times |T_j|} \tag{15}$$

$$H(P) = -\sum_{i=1}^{np}|P_i|\log\frac{|P_i|}{n} \tag{16}$$

$$H(T) = -\sum_{j=1}^{nt}|T_j|\log\frac{|T_j|}{n} \tag{17}$$

where $I(P, T)$ represents the mutual information between $P$ and $T$, $H(P)$ and $H(T)$ are the entropy of partitions.

**Adjusted Mutual Information (AMI).** AMI is an adjusted version of NMI and AMI takes into account the effects of random assignment and category imbalance. AMI is defined as follows:

$$AMI(P, T) = \frac{I(P, T) - E\{I(P, T)\}}{\max\{H(P), H(T)\} - E\{I(P, T)\}} \tag{18}$$

where $E\{I(P, T)\}$ is the expected mutual information between $P$ and $T$ under random labeling assumption.

## Acknowledgments

This work was carried out in part using hardware and/or software provided by the High-performance Computing Platform of Guangxi University.

## Author Contributions

**Conceptualization:** Wei Lan, Tongsheng Ling, Qingfeng Chen, Ruiqing Zheng, Min Li, Yi Pan.

**Data curation:** Wei Lan, Tongsheng Ling.

**Formal analysis:** Wei Lan, Tongsheng Ling, Ruiqing Zheng.

**Funding acquisition:** Wei Lan.

**Investigation:** Wei Lan, Tongsheng Ling, Qingfeng Chen.

**Methodology:** Wei Lan, Tongsheng Ling.

**Project administration:** Wei Lan, Tongsheng Ling, Min Li, Yi Pan.

**Resources:** Wei Lan, Tongsheng Ling.

**Software:** Tongsheng Ling.

**Supervision:** Min Li, Yi Pan.

**Validation:** Wei Lan, Tongsheng Ling, Ruiqing Zheng.

**Visualization:** Wei Lan, Tongsheng Ling.

**Writing – original draft:** Wei Lan, Tongsheng Ling, Qingfeng Chen.

**Writing – review & editing:** Wei Lan, Tongsheng Ling, Qingfeng Chen, Ruiqing Zheng, Min Li, Yi Pan.

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
