## [Decision Letter · Decision Letter 0]

4 Aug 2024

Dear Dr. Lan,

Thank you very much for submitting your manuscript "scMoMtF: An Interpretable Multitask Learning Framework for Single-Cell Multi-omics Data Analysis" for consideration at PLOS Computational Biology.

As with all papers reviewed by the journal, your manuscript was reviewed by members of the editorial board and by several independent reviewers. In light of the reviews (below this email), we would like to invite the resubmission of a significantly-revised version that takes into account the reviewers' comments.

The reviewers raised concerns on the lack of important information including description of datasets, details of benchmarking and evaluation metrics. The authors are expected to address the reviewers' comments in a revised version in order for this manuscript to be considered.

We cannot make any decision about publication until we have seen the revised manuscript and your response to the reviewers' comments. Your revised manuscript is also likely to be sent to reviewers for further evaluation.

Sincerely,

Xiuwei Zhang

Guest Editor

PLOS Computational Biology

Sushmita Roy

Section Editor

PLOS Computational Biology

The reviewers raised concerns on the lack of important information including description of datasets, details of benchmarking and evaluation metrics. The authors are expected to address the reviewers' comments in a revised version in order for this manuscript to be considered.

Reviewer's Responses to Questions

**Comments to the Authors:**

Reviewer #1: In the paper, the authors utilize an interpretable multitask framework (scMoMtF) for comprehensive analyzing single-cell multi-omics data. The experimental results show that scMoMtF outperforms current state-of-the-art algorithms on dimension reduction, cell classification and data simulation tasks. Overall, the manuscript is well written. However, there are still some questions needed to be addressed before the acceptance:

1.The authors should ensure that all terms used in the paper are presented with their full names upon first mention. For instance, terms like SHARE-seq should be fully defined to ensure clarity for readers who may not be familiar with the abbreviations.

2.In the figures, the first letters of words should be capitalized for consistency and professionalism. For example, in Figure 7e, ensure that all labels adhere to this formatting rule.

3.The process of calculating the indicators used in the paper, such as Adjusted Rand Index (ARI) and Normalized Mutual Information (NMI), should be explicitly shown. Providing a detailed explanation of how these indicators are computed will help readers understand the methodology and validate the results. It is recommended that the authors be able to add relevant content to ensure the reproducibility of the study.

4.The details of how the concatenate operation in Equation 3 is realized should be thoroughly explained. A comprehensive description of this process will aid in the understanding of the algorithm’s implementation. Ensuring that every step of the methodology is well-documented is essential for readers who wish to replicate or build upon this work.

5.Could the authors describe the advantages and disadvantages of the method in more detail in the discussion section and describe the future directions for improvement.

Reviewer #2: The paper presents an interpretable multitask learning framework (scMoMtF) for single-cell multi-omics data analysis. The experimental results on different tasks show that scMoMtF can produce better performance than other state-of-the-art methods. In general, it is an interesting work. However, there are several issues that need to be addressed, which are listed below:

1.As the authors mentioned, the model “during the training process of dimension reduction and cell classification tasks, the interpretability module is used to enhance this process.” Could you explain in more detail what you mean by this statement.

2.In the dimension reduction task, the authors use the clustering results of the k-means method for the corresponding metrics computation and the corresponding parameters of the method should be given for the reader's reproduction.

3.For the calculation of each quantitative indicator, the authors should give clear instructions. This can help readers understand the results more clearly and reproduce the experiment.

4.In the comparison experiments of the training efficiency of each model, could you show the training time of all the comparison experiments mentioned in the paper. This can visualize the advantages of the authors' model more.

5.There should be consistency in the descriptions in the paper; the authors give a complete description of the CITE-seq and ADT techniques, but not the SNARE-seq and SHARE-seq techniques. It is hoped that the authors will take note of such errors and correct them.

Reviewer #3: A single cell multi-omics multitask learning methods was developed in this manuscript to solve multiple tasks in single-cell multi-omics data analysis including dimension reduction, cell classification, data simulation and batch effect correction. The method contains encoder, decoder, discriminator and classification modules. The performance of this method is benchmarked to existing ones in different aspects (dimension reduction, cell classification etc.) using four existing datasets. The work flow of the method is clearly presented and results are relatively well shown in graphs. However, the scientific motivation and broad impact of the methodology is not clearly presented, the application to real data is not well summarized. Also the authors are not providing sufficient details in datasets, methods, methods evaluation and results are not adequately interpreted. There are many grammar errors. I will list details below.

(1) In methodology, the method is to model two modalities. How if the data has more than two omics datasets?

(2) No details about how the developed method scMoMtF are benchmarked to other methods. The method is benchmarked to multiple methods in each aspect (dimension reduction or cell classification or batch effect correction etc.) But there is a lack of description or introduction of each method. For example. no description of the method that was benchmarked to like SHAP (Page7, Line174)

(3) Data description of the real datasets including SNARE-seq, PBMC, SHARE-seq and CITE-seq is unclear. For example, the dimension of the SNARE-seq datasets, the evaluation platform for the gene expression or chromatin accessibility from some of the datasets. What does L1, L2, L3 cell type resolutions mean in CITE-seq dataset?

(4) Not sure what quantitative metrics are used in Figure 2 e-h for clustering performance evaluation.

(5) For dimension reduction (Figure 2), how can we tell the developed method is better from Figure 2 a-d? And more details shall be provided in results about the data dimensions after the methods are applied, for example, the proportion of biomarkers that are retained in each omics dataset.

(6) It was not described how the method can simulate cells as mentioned in P6, line 141.

(7) Page 11, line 280. What is the decision rule here for determining the real data or fake data?

(8) Page 5, line 129. What does rare cells mean and why this is important?

(9) Grammar errors. Just to list a few:

Abstract Line 4, comprehensive -comprehensively

Page 3, Line 72, modality-modalities

Page 6, Line 146, selecte-selected

Page 6, Line 158, need-needs

Page 6, Line 164, batche-batches

**Have the authors made all data and (if applicable) computational code underlying the findings in their manuscript fully available?**

Reviewer #1: Yes

Reviewer #2: None

Reviewer #3: Yes
---

## [Decision Letter · Decision Letter 1]

26 Nov 2024

Dear Dr. Lan,

We are pleased to inform you that your manuscript 'scMoMtF: An Interpretable Multitask Learning Framework for Single-Cell Multi-omics Data Analysis' has been provisionally accepted for publication in PLOS Computational Biology.

Best regards,

Xiuwei Zhang

Guest Editor

PLOS Computational Biology

Sushmita Roy

Section Editor

PLOS Computational Biology

Feilim Mac Gabhann

Editor-in-Chief

PLOS Computational Biology

Jason Papin

Editor-in-Chief

PLOS Computational Biology

Reviewer's Responses to Questions

**Comments to the Authors:**

Reviewer #1: The authors have addressed all my concerns.

Reviewer #2: All my concerns have been solved.

Reviewer #3: The authors have addressed all the concerned I had in previous round of revision.

**Have the authors made all data and (if applicable) computational code underlying the findings in their manuscript fully available?**

Reviewer #1: Yes

Reviewer #2: None

Reviewer #3: Yes

---

## [Editor Report · Acceptance letter]

3 Dec 2024

PCOMPBIOL-D-24-00810R1 

scMoMtF: An Interpretable Multitask Learning Framework for Single-Cell Multi-omics Data Analysis

Dear Dr Lan,

I am pleased to inform you that your manuscript has been formally accepted for publication in PLOS Computational Biology. Your manuscript is now with our production department and you will be notified of the publication date in due course.

With kind regards,

Anita Estes
